# A Subset of Primary Polydipsia, “Dipsogneic Diabetes Insipidus”, in Apparently Healthy People Due to Excessive Water Intake: Not Enough Light to Illuminate the Dark Tunnel

**DOI:** 10.3390/healthcare9040406

**Published:** 2021-04-01

**Authors:** Krishnaraju Venkatesan, Kumarappan Chidambaram, Premalatha Paulsamy, Ramasubbamma Ramaiah, Ali Al-Qahtani, Kumar Venkatesan, Ester Mary Pappiya, Swetha Devidi, Kalpana Krishnaraju

**Affiliations:** 1Department of Pharmacology, King Khalid University, Abha 61441, Saudi Arabia; kumarappan@kku.edu.sa (K.C.); amsfr@kku.edu.sa (A.A.-Q.); kumarve@kku.edu.sa (K.V.); 2Faculty of Nursing, King Khalid University, Abha 61441, Saudi Arabia; pponnuthai@kku.edu.sa (P.P.); rramaiah@kku.edu.sa (R.R.); 3Regional Nursing Administration, Ministry of Health, Najran 66262, Saudi Arabia; epappiya@moh.gov.sa; 4Department of Pharmacy, SMGOIH, Jawaharlal Nehru Technological University, Hyderabad 508284, India; swetha.reddy27@gmail.com; 5Erode College of Pharmacy, Tamil Nadu Dr. M.G.R. Medical University, Tamilnadu 638112, India; rnkkalpana@gmail.com

**Keywords:** dipsogenic diabetes insipidus, habitual polydipsia, polyuria-polydipsia syndrome, dipsogenic polydipsia, water intoxication, hyponatraemia

## Abstract

Dipsogenic diabetes insipidus (DDI) is a subtype of primary polydipsia (PP), which occurs mostly in healthy people without psychiatric disease. In contrast, PP is characterized by a polyuria polydipsia syndrome (PPS) associated with psychiatric illness. However, the pathogenesis of DDI is not well established and remains unexplored. In order to diagnose DDI, the patient should exhibit excessive thirst as the main symptom, in addition to no history of psychiatric illness, polyuria with low urine osmolality, and intact urine concentrating ability. Treatment options for DDI remain scarce. On this front, there have been two published case reports with successful attempts at treating DDI patients. The noteworthy commonalities in these reports are that the patient was diagnosed with frequent excessive intake of water due to a belief that drinking excess water would have pathologic benefits. It could therefore be hypothesized that the increasing trend of excessive fluid intake in people who are health conscious could also contribute to DDI. Hence, this review provides an overview of the pathophysiology, diagnosis, and treatment, with a special emphasis on habitual polydipsia and DDI.

## 1. Introduction

With the increasing acceptance of lifestyle programs and the common belief that drinking several liters of fluid per day is healthy, the prevalence of this phenomenon is increasing, particularly outside of the psychiatric setting [1,2]. This phenomenon which occurs in healthy people without psychiatric disease, seems to be closely related with dipsogenic diabetes insipidus (DDI). However, data in the overall population have not been studied and yet to be explored. DDI is defined as a rare non-psychiatric syndrome of disordered thirst, in which the osmotic threshold for thirst is abnormally low, below the threshold for antidiuretic hormone (ADH) release [3]. As DDI is not a well-established disease, there is less information available about DDI, and hence less knowledge about its prevalence in aquaholics. This review is therefore intended to provide a summary of the pathophysiology, diagnosis, and care, with a particular focus on the possibility that DDI may be triggered by habitual polydipsia.

## 2. Pathophysiology

DDI is a polyuria polydipsia syndrome (PPS) that mostly occurs in healthy people without psychiatric disease, in contrast to primary polydipsia (PP) that occurs in patients with psychiatric disease. Furthermore, as DDI occurs in healthy people without psychiatric disease, it is often confused with partial central diabetes insipidus (PCDI). Moreover, DDI is not a universally recognized PPS. DDI is caused by a disorder in the patient thirst center, where the osmotic threshold to release antidiuretic hormone is abnormally low [4]. It is a subset of PP, and first described in 1987 by Robertson G in patients without psychiatric illness [5]. This PPS has low urine osmolality, normal plasma osmolality, and normal urinary concentrating capacity. In both PP and DDI, the patient consumes an excessive amount of fluid, most frequently water, which decreases plasma osmolality followed by decreased antidiuretic hormone and less aquaporin-2 (AQP2) expression, resulting in hypotonic polyuria. This is in contrast to central diabetes insipidus (CDI), where excessively dilute urine is followed by ingestion of excess fluid [4]. In DDI, if the lower plasma osmolality is maintained for a continuous period, than the body sets this range as the threshold for ADH release [6]. Though there are discrepancies, many authors have reported that in healthy people the thirst sensation would occur at a higher osmotic threshold, compared to ADH release [7,8,9]. In DDI patients, the thirst threshold will be lower than that of ADH release [3].

Hyponatremia and the ensuing complications are the major risk factor due to excessive drinking [1]. Treatment options for DDI remain scarce. Low dose intranasal desmopressin associated with strict water restriction was reported to be useful treatment for DDI in a few studies. In this scenario, two case reports were diagnosed as DDI due to excessive fluid intake to improve health [3,10].

The etiology could be idiopathic, or due to a structural lesion similar to CDI [3]. Another hypothesis is that it could be due to behavioral aspects, due to a belief that drinking excess water will protect against the formation of nephrolithiasis or have more pathologic benefits [11,12]. Hence, as DDI patients have no signs of mental illness or increased thirst, the disorder could also be named “habitual polydipsia” [11,13].

DDI could be due to lesions in the hypothalamus and is also possible in lifestyle conscious men and women who consume an excess of water to detox the body. Excess consumption of water is also seen in sports people [14,15,16,17]. Hypothalamic lesions are primarily caused by traumatic brain injury leading to vascular or infiltrative diseases and DDI [13,18,19,20,21]. In society, constant motivation to drink more water influences some people, and such habitual polydipsia or behavioral polydipsia alters the thirst threshold and lowers it, which could lead to DDI [22,23]. PP and DDI have been reported to occur preferentially in women [24]. In general, DI thirst is qualitatively different from normal thirst, where the sensation is not only augmented but also appears to be abnormal [25].

## 3. Differential Diagnosis of DDI and Other PPS

Arginine vasopressin (AVP), a antidiuretic hormone (ADH), and thirst are the two factors that maintain water homeostasis. If these two regulatory mechanisms face any disturbance then it leads to PPS, which includes three different conditions: CDI due to insufficient secretion of AVP, nephrogenic DI caused by renal insensitivity to AVP action, and PP/DDI due to excessive fluid intake and a consequent physiological suppression of AVP [26].

AVP is a peptide that consists of nine amino acids, and is derived from a precursor protein that has 164 amino acids. The precursor protein consists of a signal peptide, AVP itself, neurophysin-2, and the c-terminal glycoprotein, copeptin [27]. AVP is synthesized in the hypothalamus and stored in the posterior pituitary. When there is increase in serum osmolality from the normal threshold value of 280 mOsm/kg, a linear increase in AVP is noted [28,29]. The released AVP binds to vasopressin-2 receptors on the renal collecting tubular cell, leading to water retention and an increase in urinary concentration. The other mechanism that has an impact on water homeostasis is thirst. Increase in serum osmolality leads to an increase in thirst sensation [29]. The physiological mechanism by which thirst occurs is not clear, but osmoreceptors in the hypothalamus seems to have a role [29,30,31].

The first step in evaluating patients with PPS is to verify that there is polyuria and that it is hypotonic in nature. For polyuria, the urine output should be in excess of 40 mL/kg/24 h, and to confirm PPS, the urine osmolality should be <300 mOsm/kg. If the urine osmolality is above this value, then other etiologies that are not primarily caused by impaired AVP secretion or action have to be ruled out, such as hyperglycaemia or hypercalcaemia. [32,33,34] Once hypotonic polyuria is confirmed, history has to be taken into account to characterize PPS. This includes onset of polyuria, whether it is gradual or abrupt, craving for ice water, and family history to rule out hereditary DI (diabetes insipidus) [35].

In order to diagnose DDI, the patient should exhibit excessive thirst as the main symptom intact and urine concentrating ability. Moreover, as DDI could be possible due to the similar hypothalamic lesions of CDI (Table 1), c-MRI scans could be done to diagnose PP as an idiopathic or psychiatric based illness [36].

Since the introduction of a test protocol by Miller in 1970 [37], a water deprivation test (Table 2) followed by administration of desmopressin is the standard method used for confirmation and differentiation of various types of PPS, including PP. He studied the response to water deprivation tests in PPS patients, but lacked data about DDI patients. Still, this study and other related PP sources are helpful as the main differentiating factor among DDI and PP is psychiatric etiology, and the rest are almost same.

Miller made a clinical study involving healthy volunteers and patients with PPS, where they underwent fluid deprivation for 8 h followed by desmopressin administration. The urine osmolality rose >800 mOsm/kg for healthy volunteers (*n* = 10), and did not increase above plasma osmolality for central and nephrogenic DI. Whereas, patients with PCDI and PP showed similar urine concentrating ability above plasma osmolality. This highlights the difficulty in differentiating PCDI and DDI. An additional clue reported to be useful in differentiating PCDI and DDI is sodium level in the bloodstream, where normonatremia is noted for DDI as opposed to mild hypernatremia in PCDI.

However, a recent prospective observational study reported on patients (*n* = 23, 74% female) with increased fluid consumption leading to hyponatremia and being admitted to an emergency setting due to PP. Notably, one third of the patients had no psychiatric complaints and were categorized into the DDI category. This was the first study to report hyponatremia in DDI patients. Though it is not common in DDI patients, acute illness associated with AVP stimulating stress may induce hyponatremia [38]. To support this, there have been case reports in infectious DDI patients who developed hyponatremia [1,39]. Hence, the author suggested rationalizing the general popular advice “drink a lot in acute illness” to avoid increased fluid consumption leading to DDI and associated hyponatremia.

In the Miller study, after desmopressin administration, patients with central and nephrogenic DI showed around a 50% increase in urine osmolality, whereas PCDI, PP, and healthy volunteers showed >9%, 2% to 3%, and 5%, respectively. From the above data, there seems to be clarity in differentiating diagnosis among PPS. However, the difference appears minimal when differentiating PCDI and PP. The unique reported difference is patients with PP had a urine osmolality increased to >10% compared to CDI, with urine osmolality >800 mOsm/kg at the end of water deprivation period [37].

Currently, in many endocrine centers the water deprivation test is used for diagnosis of PPS. However, this test has some lacuna, as the prolonged water deprivation not only creates discomfort to patients, leading to discontinuation [26], it also creates resistance to desmopressin [40], due to downregulation of aquaporin-2 synthesis. This desmopressin resistance diminishes the diagnostic distinction between PCDI, partial nephrogenic DI, and PP, as all three respond slightly to desmopressin after water deprivation [36,37].

As an improvement over the water deprivation test, in 1980 Zerbe et al. studied the role of plasma AVP measurement [41]. It was found to have role in differentiating PCDI, DDI, and nephrogenic DI. For three reasons it does not seem to improve the PPS diagnosis. First, AVP is a labile peptide with a half-life of 6 min and the associated possible errors in processing [42]. Second, it is mostly bound to platelets, and hence platelet rich samples show high AVP values [43]. Third, a large cohort study in healthy volunteers has shown that the relationship between AVP and plasma osmolality is non-linear [44]. In addition, regardless of the etiology of PPS, polyuria itself can trigger abnormality in urine concentrating ability due to loss of medullary tonicity and downregulate AVP release. This has been reported in PP patients under osmotic stress [44]. Hence, for differential diagnosis, AVP measurement has not been put into practice.

In addition to the above mentioned routine analysis, Fenske et al. suggested determination of copeptin, a precursor peptide of AVP and mirrors circulating AVP concentration [44,45]. Compared to AVP, copeptin can be easily measured, is more stable in plasma and serum, and has a similar half-life [46]. In addition, in two studies (*n* = 50 and 55) [44,47] where differential diagnosis of PPS was done using copeptin, the sensitivity and specificity ranged from 82–100% and 92–100%. The only concern with copeptin is that its level increases in acutely ill patients and may be misleading in patients with PPS, and hence should be interpreted with caution [48]. The plasma level of another neuro-vasoactive peptide apelin was reported to be helpful in the differential diagnosis of PPS. As the reported data was from a proof of concept study, larger studies are required for confirmative correlation of apelin in PPS [49]. Hence based on the above data, for the differential diagnosis of PPS, the water deprivation test in combination with copeptin measurement may become a standard test in the future.

## 4. Treatment

Antidiuretic therapy was the early treatment for DDI. Due to the risk of water intoxication, its usage has not been in practice. In 1990, Ferrer et al. reported a DDI case in a 32-year-old woman who had PPS and had administered intranasal desmopressin 48 h prior to admission to his hospital. She developed headaches, drowsiness, unsteadiness, vomiting, and suffered generalized tonic-clonic seizures. Hence, he reported that intranasal desmopressin should be avoided in patients with DDI [10].

In contrast to this report, in 2006, a DDI case has been reported in a 26-year-old healthy woman without psychiatric illness referred for PPS. A hypertonic saline infusion test was done to measure urine osmolality, plasma osmolality, serum sodium, and plasma ADH levels. To measure the osmotic threshold for thirst, a modified water deprivation test was done, in addition to a hypotonic water infusion test. Through this test, and based on body weight, total body water volume, and plasma sodium, the approximate required intake of water was determined. Her clinical findings included, eunatremic, intact urine concentrating ability, and polyuric with low urine osmolality. Most interestingly, the patient’s osmotic threshold for thirst was found to be shifted downward, occurring at a lower set point than the osmotic threshold for ADH release, and consistent with the diagnosis of DDI. The physician successfully treated this DDI with low dose intranasal desmopressin (2.5 μg, intranasally). Low dose intranasal desmopressin was initiated with strict water restriction, as per the required amount. She showed significant results in control of polyuria and functional improvement [3]. Water restriction during desmopressin treatment is what helped this patient to recover, contrary to the earlier reported case. The author also suggested that retraining of the osmotic thirst threshold, perhaps through biofeedback, seems to have a role in DDI treatment. Its impact on ADH release and other associated abnormalities needs to be explored [3].

In 2011, an animal study showed that GLP-1R (Glucagon-like peptide-1 receptor) agonists have a hypodipsic effect through central activation of GLP-1R [50]. Hence, the potential of GLP-1R agonists like liraglutide and dulaglutide in the treatment of DDI has to be explored. In 2012, Shapiro M et al. suggested that sour candies, ice chips, or gum might be useful, if used as alternatives to fluid intake [35]. A DDI case study reported that in 2018 a 19-year-old male was successfully treated with low dose desmopressin in addition to fluid restrictions [51].

## 5. Could Excessive Drinking Habits Lead to DDI Associated with Water Intoxication?

Water is vital to the body as it maintains body temperature, excretion, and all major body functions [52]. Additionally, it makes up two thirds of body weight, and constantly is lost through sweat, urination and breathing. Symptoms of dehydration are detectable through thirst and we drink water. If there is less fluid consumption, it might lead to cardiometabolic diseases, recurrent kidney stones, and chronic kidney disease. In addition, adequate hydration helps to improve cognition in children and adults [53,54,55,56,57,58]. However, when we drink more water, it dilutes sodium and other electrolytes leading to water intoxication. This initially leads to headache, nausea, and vomiting, while severe cases can lead to serious conditions like drowsiness, muscle cramping, double vision, seizure, coma, or even death. Moreover, excess consumption of water leads to depletion of potassium, leading to symptoms like leg pain, irritation, and chest pain, as it plays a major role in the contraction and relaxation of muscles. Even in the case of sports people, to avoid exercise induced hyponatremia, it is advised to drink water only when they are thirsty [52]. Notably, if they are consuming 900 mL of fluid over a one hour period, it reduces the speed and duration of exercise [59]. This is due to the diversion of blood from working muscles towards the gastrointestinal tract due to its distension following high fluid consumption [60].

There is a strong belief in society that drinking excessive water will purify and cleanse the body by removing harmful waste products. In addition, there are old unsubstantiated claims like the 8 × 8 rule (8 × 250 mL of water/day) that have made people believe that feeling thirsty means that we are already dangerously dehydrated. In addition, even scientific studies done on different disease conditions tend to support high fluid intake. A 10-year cohort study (*n* = 48000 men) done in health professionals reported that fluid consumption is inversely proportional to the risk of bladder cancer [61]. Similar results have been reported for colon cancer, chronic kidney disease, urolithiasis, urinary tract infection, fatal coronary heart disease, venous thromboembolism, and exercise-induced bronchospasm [62,63,64,65,66,67,68,69,70,71]. However, experts believe that we do not require fluid in excess of the amount our body actually signals for. Adding to it, if thirst is an imperfect signal of fluid deficit, than how come everything else in the body could be perfect? Moreover, it has worked well since the evolution of human race. Actually, the thirst sensation is triggered when the body loses water by 1–2%, and during which physical and cognitive performance might also decline [72,73,74,75,76,77].

Though studies have shown that drinking enough to avoid mild dehydration helps to support brain function, it lacks strong benefits beyond the point of avoiding dehydration. Though there is less evidence to support that the habit of consuming excess fluid could lead to DDI, the available scientific data to support such a claim cannot be ignored. Recently, a DDI case was reported in the UK in a 47-year-old women with hyponatremia [2]. She had consumed excess fluid as part of a new year “detox” concept, along with some natural products. A similar case was reported earlier in the UK in a 40-year-old man with a history of generalized anxiety disorder, who similarly presented with seizures due to severe hyponatremia [78]. In both cases, though there was the involvement of a herbal product, excessive fluid intake was also involved. In addition, there seems to be a considerable percentage of people in society who generally consume excess fluid. A secondary analysis study (*n* = 3214) done in six countries (Argentina, Brazil, China, Indonesia, Mexico, and Uruguay) showed that 11% (*n* = 352) were high drinkers and 8% (*n* = 264) were very high drinkers of water [79].

Moreover, anorexia nervosa, which was earlier described as a psychiatric disorder, is now reported as a behavioral disorder [80]. There are reports in patients with eating disorders with serious complications due to hyponatremia, which is caused by excessive fluid intake to maintain less weight [81]. As the disorder is behavioral, it could be considered as DDI associated with an eating disorder.

## 6. Conclusions

In conclusion, DDI is a little studied and reported disease. As DDI cases have been reported in health-conscious people consuming excess fluid, it could be hypothesized that habitual polydipsia might be a cause of DDI. To substantiate this claim, more detailed studies in habitual over drinkers needs to be done. Moreover, specific research investigating novel treatment strategies is desirable. Most importantly, enhancing the awareness in the general public to rationalize the common belief that drinking excessive water will purify and cleanse the body is the need of the hour.

## 7. Summary

✓Dipsogenic diabetes insipidus (DDI) is a subtype of primary polydipsia (PP) and occurs mostly in healthy people without psychiatric disease.✓The pathogenesis of DDI is not well established and remains unexplored.✓Most interestingly, the patient osmotic threshold for thirst was found to be shifted downward, occurring at a lower set point than the osmotic threshold for ADH release.✓The constant motivation in society to drink more water influences some people and such habitual polydipsia or behavioral polydipsia alters the thirst threshold and lowers it. It is hypothesized that this scenario could lead to DDI.✓Currently in many endocrine centers, the water deprivation test is the standard used for the diagnosis of PPS.✓Low dose desmopressin in addition to fluid restriction seems to be a promising treatment for DDI.✓There is less knowledge about DDI and limited studies have been available until now to support the hypothesis that the habit of consuming excess fluid could lead to DDI. Nevertheless, the available scientific data supporting such a claim cannot be ignored.

## 8. Future Directions

✓For the differential diagnosis of polyuria polydipsia syndrome, copeptin measurement, in addition to water deprivation testing, seem to have more value and have to be explored.✓For treatment of DDI, retraining of the osmotic thirst threshold through biofeedback and its impact on ADH release and other associated abnormalities needs to be explored.✓To identify an answer to this intriguing hypothesis that DDI might be possible in people with habitual consumption of excess fluid, well-designed larger observational studies in healthy people having the habit of consuming excess fluid are warranted.

## Figures and Tables

**Table 1 healthcare-09-00406-t001:** Etiologies to consider in the differential diagnosis of polyuria polydipsia syndrome (PPS). DI: diabetes insipidus.

Type of DI	Sub-Type of DI	Etiologies
Central diabetes insipidus	Acquired	✓ Trauma (surgery, deceleration injury) ✓ Vascular (cerebral hemorrhage, infarction anterior communicating artery aneurysm or ligation, intrahypothalamic hemorrhage) ✓ Neoplastic (craniopharyngioma, meningioma, germinoma, pituitary tumor, or metastases) ✓ Granulomatous (histiocytosis, sarcoidosis) ✓ Infectious (meningitis, encephalitis) ✓ Inflammatory/autoimmune (lymphocytic infundibuloneurohypophysitis) ✓ Drug/toxin-induced (ethanol, diphenylhydantoin, snake venom) ✓ Other disorders (hydrocephalus, ventricular/suprasellar cyst, trauma, degenerative diseases) ✓ Idiopathic ✓ Congenital malformations ✓ Autosomal dominant: AVP-neurophysin gene mutations
	Congenital	✓ Autosomal recessive: Wolfram Syndrome ✓ X-linked recessive ✓ Idiopathic
Nephrogenic diabetes insipidus	Factors	✓ Drug-induced: demeclocycline, lithium, cisplatin, methoxyflurane, etc. ✓ Metabolic disease: hypercalcemia, hypokalemia ✓ Systemic disorders: Infiltrating lesions (sarcoidosis, amyloidosis, multiple myeloma, Sjoergen’s disease) ✓ Vascular: sickle cell disease ✓ Chronic renal disease: polycystic kidneys, obstructive uropathy ✓ Osmotic diuretics: glucose, mannitol ✓ Pregnancy:
	Idiopathic	
Dipsogenic (downward resetting of thirst threshold)		
		✓ Compulsive water drinking ✓ Associated with affective disorders ✓ Drug induced ✓ Structural/organic hypothalamic disease ✓ Sarcoidosis ✓ Tumors involving hypothalamus ✓ Head injury ✓ Tuberculous meningitis

**Table 2 healthcare-09-00406-t002:** Protocol for water deprivation/desmopressin test.

Phases of Water Deprivation Test	Parameters
Initial tests and preparation	✓ Polyuria should be confirmed by 24 h urine volume ✓ Rule out diabetes mellitus, Urinary tract Infection, hypercalcaemia, hypokalaemia, renal failure and thyrotoxicosis ✓ Morning urine osmolality >600 mOsm/kg rules out DI and therefore a water deprivation test is not necessary ✓ Free access to fluid overnight prior to test ✓ No alcohol or caffeine ✓ 7am light breakfast but NO fluids, tea, coffee, or smoking
Dehydration phase	✓ Restrict fluids for 8 h ✓ Weigh patient at 2 hourly intervals ✓ Plasma and urine osmolality, and urine volume measurements 2 hourly ✓ Stop test if weight loss exceeds 5% of starting weight, or thirst is intolerable ✓ Supervise patient closely to avoid non-disclosed drinking
Desmopressin phase	✓ Inject intramuscularly 1 mcg desmopressin ✓ The patient is allowed to eat and drink, even up to 1.5–2 times the volume of urine passed during the dehydration phase. ✓ Urine output, urine osmolality, serum sodium, and plasma osmolality are measured hourly for 1–2 h after desmopressin administration

## Data Availability

No new data were created or analyzed in this study. Data sharing is not applicable to this article.

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
