# Peer review of "A Subset of Primary Polydipsia, “Dipsogneic Diabetes Insipidus”, in Apparently Healthy People Due to Excessive Water Intake: Not Enough Light to Illuminate the Dark Tunnel"

_healthcare, 2021, doi:10.3390/healthcare9040406_

Round 1

Reviewer 1 Report

General Comment: This is an interesting review of a relatively under-recognized clinical condition.

Major Comments:

  1. The authors use a lot of abbreviations,such as PP, PPS, DDI, etc.  This makes the review hard to read, especially as these conditions are uncommon and not known to a reader.
  2. Line 81 - nephrogenic CDI is incorrect - it should be nephrogenic diabetes insipidus (NDI)
  3. Table 2 - under initial tests, it says diabetes.  might be better to say diabetes mellitus to distinguish from diabetes insipidus
  4. Table 2 - under desmopressin phase - why is desmopressin given IM rather than intra-nasal?
  5. Lines 171 and 173 - inrranasal should be intranasal

Author Response

Dear Reviewer,

Thank you very much for your valuable comments and feedback regarding our research paper. Please find below our response.

  1. I agree with reviewer comments. In the initial draft, we used the full form for all medical terms. But, as the medical term for which we used abbreviations are lengthy and more terms are repeatedly used, we changed to abbreviations 
  2.  Changed as per reviewer comments
  3. Changed as per reviewer comments
  4. IM route is mentioned based on the summary of product characteristics (SPC) of desmopressin (https://www.medicines.org.uk/emc/product/5447/smpc#gref )
  5. Modified as per reviewer comments 

Reviewer 2 Report

The concept of dipsogenic diabetes insipidus (DDI) is not well established. This reviewer thinks that DDI is one form of primary polydipsia or that a condition the authors would like to call DDI is 'primary polydipsia (PP) without psychiatric diseases'.

If the authors would like to use the term, DDI, they should clearly define this condition in this manuscript and need to explain why this condition should be defined.

In the light of the concern that DDI is not a well-defined entity, the sentence like 'excessive drinking habit lead to DDI is misleading. This can be just simply replaced by 'excessive drinking habit could lead to hyponatremia associated with water intoxication.

This reviewer suggest hyponatremia induced by excessive water intake in apparently healthy people should be more focused in this manuscript, and therefore the title of 'hyponatremia induced by excessive water intake in apparently healthy people (something like this)' may be used.

(Minor points)

DDI in L36 and PPS in L40 should be full-spelled, since those are first used in the main context.

Table 1 ; In this table, the authors apparently divide PP into two conditions, psychogenic PP and DDI. This is not explained in the manuscript. Since PP is not a form DI, 'Type of DI' in the first column is confusing.

'4.Treatment' includes a lot of comments on differential diagnosis.

Are there any papers published to prove/quantify the low osmotic threshold of the thirst in patients who were claimed to have DDI?

Author Response

Dear Reviewer

Thanks for your highly valuable comments and feedback regarding our research paper

Please find below Our response:

  1. We agree with the reviewer comments that concept of DDI is not well established. At the same time, DDI is designated as an orphan disease by both Europe and USFDA regulatory and other authorities. Hence, we intended to define this term in our article.
  2. We agree with reviewer comments that DDI is not a well-defined entity, and hence the sentence like 'excessive drinking habit lead to DDI is misleading.  So as per reviewer comments, we modified as  'excessive drinking habit could lead to DDI associated with water intoxication
  3. As per valuable comments from reviewer, we modified the title "Dipsogenic Diabetic Insipidus in apparently healthy people due to excessive water intake: Not enough light to illuminate the dark tunnel"
  4. We agree with reviewer comments and made modifications in Table 1, under Type of DI

Round 2

Reviewer 2 Report

Major comments;

  1. The authors should state clearly in both Abstract and Introduction that DDI is not well established or widely known.
  2. Is DDI a subtype of primary polydipsia (PP) or not? This reviewer thinks that it is a subtype of PP, which should be stressed in the manuscript (since the term DDI, which includes DI in the word order, can be misleading for the beginners). In this sense, this reviewer suggests that ‘A subset of primary polydipsia, ‘’dipsogneic diabetes insipidus’’, in apparently healthy people due to excessive water intake as the title of the manuscript.
  3. It is difficult to study the osmotic thirst threshold. Therefore, how many papers convincingly analyzed the threshold in patients who were claimed to have DDI. This issue should be described more concretely in the manuscript.
  4. The authors have not defined DDI yet in the revised manuscript. What they do is just to overview the disease entity. The one-two key sentences like DDI is defined by---- is enough.
  5. DDAVP use in the treatment for PP/DDI should be discouraged even at a low dosage. Water restriction is the main stream in the treatment. The treatment of both DDAVP and water restriction is contradictory. If you would like to refer the previous related case reports on the use of DDAVP, the dose should be precisely described for the reader to understand how low it is.
  6. Summary could more focus on ‘DDI’.

Minor comments;

  1. L34-35; With the increasing----- is healthy

  Relevant references are needed to support this phrase.

  1. L48; you state that it (DDI) is a subset of PP, whereas you started with ‘In both PP and DDI---‘. This is confusing.
  2. L68; the word iatrogenic is erroneously used.
  3. L81; the sentence that ADH also known as AVP is not accurate scientifically. AVP is the substance itself which has a role in antidiuresis, whereas ADH has a broader meaning.
  4. L112; what would you like to say in the phrase starting with ‘but does not ---‘
  5. L123; PCDI should be full-spelled.
  6. L135; miller should be capitalized.

Author Response

Manuscript ID: healthcare-1162571 

Article Title:

A subset of primary polydipsia, ‘’dipsogneic diabetes insipidus’’, in apparently healthy people due to excessive water intake: Not enough light to illuminate the dark tunnel

Reviewer-2:

Major comments;

  1. The authors should state clearly in both Abstract and Introduction that DDI is not well established or widely known.

Response:

 As per reviewer comments, the statement that “DDI is not well established or widely known”” has been captured in both abstract and Introduction. Line 20 and 36-42.

2. ”Is DDI a subtype of primary polydipsia (PP) or not? This reviewer thinks that it is a subtype of PP, which should be stressed in the manuscript (since the term DDI, which includes DI in the word order, can be misleading for the beginners). In this sense, this reviewer suggests that ‘A subset of primary polydipsia, ‘’dipsogneic diabetes insipidus’’, in apparently healthy people due to excessive water intake as the title of the manuscript.

Response:

We agree with reviewer comments about the title, and the title has been modified. Line 1 – 4.

3. It is difficult to study the osmotic thirst threshold. Therefore, how many papers convincingly analyzed the threshold in patients who were claimed to have DDI. This issue should be described more concretely in the manuscript.

Response:

 As per reviewer comments, points relevant to measuring osmotic thirst threshold in DDI patients were stated clearly in the article. Line 183-189.

4. The authors have not defined DDI yet in the revised manuscript. What they do is just to overview the disease entity. The one-two key sentences like DDI is defined by---- is enough.

 Response:

  As per reviewer comments, definition has been included. Page 39-41.

5. DDAVP use in the treatment for PP/DDI should be discouraged even at a low dosage. Water restriction is the main stream in the treatment. The treatment of both DDAVP and water v restriction is contradictory. If you would like to refer the previous related case reports on the use of DDAVP, the dose should be precisely described for the reader to understand how low it is.

Response:

  As per reviewer comments, Dose of DDAVP was included. Line 193.

6. Summary could more focus on ‘DDI’.

Response:

As per reviewer comments, the summary is modified. Line 266-273.

Minor comments;

  1. L34-35; With the increasing----- is healthy

             Relevant references are needed to support this phrase.

             Response:

   Reference added as per reviewer comments. Line 34-36

2. L48; you state that it (DDI) is a subset of PP, whereas you started with ‘In both PP and DDI---‘. This is confusing.

Response:

We understand the concern of the reviewer. We have used “in both PP and DDI”” in one sentence, to differentiate PP/DDI from CDI. Line 55.

3. L68; the word iatrogenic is erroneously used.

Response:

We have modified as per reviewer comments. Line 72.

4. L81; the sentence that ADH also known as AVP is not accurate scientifically. AVP is the antidiuresis, whereas ADH has a broader meaning substance itself which has a role in

Response:

We have modified as per reviewer comments. Line 85.

5. L112; what would you like to say in the phrase starting with ‘but does not ---‘

Response:

 We have modified as per reviewer comments. Line 115-116.

6. L123; PCDI should be full-spelled.

Response:

 As we have full spelled in Line 50, we have mentioned as PCDI

7. L135; miller should be capitalized.

Response:

 We have modified as per reviewer comments. Line 139.